# 7-Ketocholesterol Promotes Retinal Pigment Epithelium Senescence and Fibrosis of Choroidal Neovascularization via IQGAP1 Phosphorylation-Dependent Signaling

**DOI:** 10.3390/ijms241210276

**Published:** 2023-06-17

**Authors:** Haibo Wang, Aniket Ramshekar, Thaonhi Cung, Chris Wallace-Carrete, Chandler Zaugg, Jasmine Nguyen, Gregory J. Stoddard, M. Elizabeth Hartnett

**Affiliations:** 1John A. Moran Eye Center, University of Utah, Salt Lake City, UT 84132, USA; hwang6@lsuhsc.edu (H.W.); aniket.ramshekar@hsc.utah.edu (A.R.); thaonhi.cung@ttuhsc.edu (T.C.); chris.wallacecarrete@hsc.utah.edu (C.W.-C.); chandler.zaugg@utah.edu (C.Z.); jasmine.nguyen@utah.edu (J.N.); 2Department of Pathology, LSU Health Sciences Center, New Orleans, LA 70112, USA; 3Department of Internal Medicine, University of Utah, Salt Lake City, UT 84132, USA; greg.stoddard@hsc.utah.edu; 4Byers Eye Institute, Stanford University, Palo Alto, CA 94304, USA

**Keywords:** retinal pigment epithelium, cellular senescence, mammalian target of rapamycin (mTOR), macular degeneration, IQ motif containing GTPase activating protein (IQGAP1)

## Abstract

Accumulation of 7-ketocholesterol (7KC) occurs in age-related macular degeneration (AMD) and was found previously to promote fibrosis, an untreatable cause of vision loss, partly through induction of endothelial-mesenchymal transition. To address the hypothesis that 7KC causes mesenchymal transition of retinal pigment epithelial cells (RPE), we exposed human primary RPE (hRPE) to 7KC or a control. 7KC-treated hRPE did not manifest increased mesenchymal markers, but instead maintained RPE-specific proteins and exhibited signs of senescence with increased serine phosphorylation of histone H3, serine/threonine phosphorylation of mammalian target of rapamycin (p-mTOR), p16 and p21, β-galactosidase labeling, and reduced LaminB1, suggesting senescence. The cells also developed senescence-associated secretory phenotype (SASP) determined by increased IL-1β, IL-6, and VEGF through mTOR-mediated NF-κB signaling, and reduced barrier integrity that was restored by the mTOR inhibitor, rapamycin. 7KC-induced p21, VEGF, and IL-1β were inhibited by an inhibitor of protein kinase C. The kinase regulates IQGAP1 serine phosphorylation. Furthermore, after 7KC injection and laser-induced injury, mice with an IQGAP1 serine 1441-point mutation had significantly reduced fibrosis compared to littermate control mice. Our results provide evidence that age-related accumulation of 7KC in drusen mediates senescence and SASP in RPE, and IQGAP1 serine phosphorylation is important in causing fibrosis in AMD.

## 1. Introduction

Age-related macular degeneration (AMD) is a leading cause of vision loss worldwide and is characterized by drusen, vision-threatening macular neovascularization, and atrophic and outer retinal degeneration [1]. Drusen develop beneath the basal aspect of the retinal pigment epithelial cell (RPE) monolayer and are composed of many substances, including complement, cholesterol, and lipoproteins [2,3]. A prominent substance measured in drusen is 7-ketocholesterol (7KC) [4], an oxidized form of cholesterol and cholesterol fatty acid esters. Although 7KC can increase in the bloodstream with increasing age [5], it is also produced in the retina by the oxidation of accumulated cholesterol [6]. The pathologic effects of 7KC include inflammation and cytotoxicity [7] and have been extensively studied in cardiovascular and neurodegenerative diseases [5,8] but less studied in ocular diseases, such as AMD. 

Evidence links 7KC with AMD [9,10,11]. Compared to younger donor eyes, 7KC levels in RPE-associated drusen were significantly higher in older donor eyes [4]. 7KC chemoattracts retinal microglia [9], which causes choroidal neovascularization in mouse models [11,12]. 7KC-mediated microglial activation and migration involved pyrin domain containing receptor 3 (NLRP3) inflammasome activation with the release of proinflammatory factors, interleukin-1 beta (IL-1β), interleukin-6 (IL-6), interleukin-18 (IL-18), and tumor necrosis factor alpha (TNFα) [9]. In cultured ARPE-19 cells, 7KC induced an inflammatory response involving toll-like receptor 4 (TLR4)-mediated endoplasmic reticulum stress [13]. 7KC was also reported to induce apoptosis through caspase-dependent signaling in ARPE-19 [10]. Counteracting the effects of 7KC in several age-related diseases has been considered, including through neutraceutical intervention [14].

Since 7KC accumulates with aging and in AMD, we were interested in the effects of 7KC on RPE and choroidal endothelial cells (CECs), two cell types important in the pathology of advanced forms of AMD. We previously reported that 7KC altered the phenotype of CECs, making them mesenchymal-like, and led to increased fibrosis in mouse models of laser-induced neovascular lesions [15]. Epithelial-mesenchymal transition (EMT) in RPE has been reported in subretinal fibrosis in AMD [16]. Therefore, we formed the hypothesis that 7KC promotes EMT of the RPE. In contrast to causing EMT, we found that 7KC, at the same concentration that caused endothelial-mesenchymal transition in CECs, instead increased senescent markers, p21 and p16, reduced Lamin B1, a senescence-associated biomarker [17], and led to mammalian target of rapamycin (mTOR)-mediated senescence-associated secretory phenotype (SASP) with reduced RPE barrier integrity. Furthermore, 7KC treatment induced serine phosphorylation of the multidomain protein, an IQ motif containing GTPase activating protein (IQGAP1). Inhibition of protein kinase C (PKC) reduced 7KC-mediated IQGAP1 phosphorylation and 7KC-induced mTOR activation, senescence, and SASP of the RPE. In a laser-induced choroidal neovascularization model, mice with a serine mutation in IQGAP1 had significantly reduced 7KC-mediated alpha smooth muscle actin (αSMA)-labeled fibrotic lesions after laser-induced injury compared to littermate control mice. Together our findings support a hypothesis that age-related accumulation of 7KC in drusen mediates senescence and SASP in RPE and IQGAP1 serine phosphorylation in RPE, which is important in causing fibrosis, an untreatable cause of blindness, in AMD.

## 2. Results

### 2.1. 7-Ketocholesterol (7KC) Does Not Increase Mesenchymal Transition or Death of hRPE

We previously found that CECs underwent endothelial-mesenchymal transition after exposure to 7KC and that intravitreal 7KC increased fibrosis in laser-induced lesions in vivo [15]. To determine if 7KC led to EMT of other cells, including RPE, we measured mesenchymal cell markers, fibroblast activation protein (FAP), and alpha-smooth muscle actin (αSMA), in western blots in HPBCD- or 7KC-treated hRPE after 48 h of exposure. Following treatment, 7KC did not increase FAP or αSMA compared to HPBCD in hRPE (Figure 1A). In the same cell lysates, RPE-specific proteins, BEST1, RPE65, CRALBP, PDGFRβ, and integrin β4 [18] were not affected by 7KC treatment (Figure 1B). Together, these data suggest 7KC does not induce EMT of the RPE at the same concentration and exposure times. 7KC induces several pathophysiological effects, including inflammation, oxidative stress, and cell death [5]. To determine if 7KC led to caspase-mediated RPE death, caspase 1 and caspase 3 were measured in 7KC- or HPBCD-treated hRPE. After exposure to either treatment, cleaved caspase 3 (Appendix A) was not detected. Neither total caspase 3 (Appendix A) nor caspase 1 (Figure 1C,D) was increased after 7KC treatment. Furthermore, 7KC did not induce hRPE death determined by the number of TUNEL-positive cells (Figure 1E) or by the tetrazolium bromide (MTT) assay (Figure 1F), providing support that the concentration of 7KC used was not cytotoxic to hRPE. Rather, 7KC significantly increased the anti-apoptotic protein BCL-2 (Figure 1G) and the mRNA expression of BCL-xL (Figure 1H), which suggested that 7KC-treated hRPE exhibited an anti-apoptotic phenotype.

### 2.2. 7KC Induces Cell Cycle Arrest at Mitosis and Promotes Senescence and SASP in hRPE

One of the events involved in resistance to apoptosis is the induction of cell cycle arrest. We detected the phosphorylation of serine 10 in histone H3 (p-histone H3 (Ser 10)) in cell lysates from 7KC-treated but not control-treated hRPE (Figure 2A). This elevated p-histone H3 (Ser 10) aligned with the thinking that 7KC induced cell cycle arrest at mitosis. Since senescent cells exhibit cell cycle arrest [19], we investigated whether 7KC induced senescence of hRPE. Cellular senescence markers p16, p21, and Lamin B1 [20] were measured by western blots of hRPE treated with 7KC or HPBCD control. Compared to HPBCD, 7KC-treated hRPE had significantly increased p16 (Figure 2A,B) and p21 (Figure 2A,C), and decreased Lamin B1 (Figure 2A,D). In addition, beta-galactosidase staining was greater in 7KC compared to HPBCD-treated cells (Figure 2E,F). These results support the postulate that 7KC promotes the senescence of RPE. 

Senescent cells exhibit a complex pro-inflammatory response, known as senescence-associated secretory phenotype (SASP) [20]. SASP is regulated by the activation of nuclear factor kappa B (NF-κB) to produce pro-inflammatory factors, cytokines, and growth factors [21,22]. To determine if 7KC-induced senescent hRPE exhibited SASP, phosphorylated NF-κB (p-NF-κB), total NF-κB, IL-1β, IL-6, and VEGF were measured in hRPE treated with 7KC or HPBCD. Compared to HPBCD, 7KC-treated hRPE had significantly increased p-NF-κB/NF-κB (Figure 2G,H), IL-1β (Figure 2I,J), IL-6 (Figure 2I,K), and VEGF (Figure 2I,L). Taken together, the results support the idea that 7KC promoted hRPE senescence and SASP, characterized by cell cycle arrest, increased p16 and p21, decreased Lamin B1, and the production of pro-inflammatory and angiogenic factors.

### 2.3. Inhibition of mTOR Signaling Prevents 7KC-Mediated Senescence and SASP in hRPE

Activation of mTOR signaling is implicated in mammalian aging and has been shown to be involved in age-associated changes in human RPE [23]. Inhibition of S6K1, the mTORC1 downstream effector, extends the lifespan in mammals [24]. To determine if activation of mTOR and S6K1 were involved in 7KC-mediated RPE senescence, phosphorylated mTOR (p-mTOR) and S6K1 (p-S6K1) were measured in hRPE treated with 7KC or HPBCD for 24 h. Compared to HPBCD, 7KC-treated hRPE had increased p-mTOR and p-S6K1 (Figure 3A) as well as p-histone H3 (Ser10) (Figure 3B), p16 (Figure 3B,C), and p21 (Figure 3B,D). These proteins were also measured in hRPE pretreated with the mTOR inhibitor, rapamycin, prior to incubation with 7KC or HPBCD. Pretreatment with rapamycin efficiently inhibited p-mTOR and p-S6K1 (Figure 3A) in both HPBCD- and 7KC-treated hRPE. In the same cell lysates, compared to vehicle control, pretreatment with rapamycin inhibited 7KC-induced p-histone H3 (Ser10) (Figure 3B). Rapamycin significantly reduced p16 (Figure 3B,C) and p21 (Figure 3B,D) in both HPBCD- and 7KC-treated hRPE. These results provide evidence that activation of mTOR/S6K1 signaling was involved in 7KC-induced senescence of the RPE.

To determine if inhibition of mTOR-regulated SASP, p-NF-κB, IL-1β, IL-6, and VEGF were measured in hRPE pretreated with rapamycin or in the control prior to incubation with 7KC or HPBCD. Compared to the control, inhibition of mTOR by rapamycin inhibited 7KC-mediated p-NF-κB (Figure 3E), IL-1β (Figure 3F,G), VEGF (Figure 3F,H), and IL-6 (Figure 3F,I). Pretreatment with rapamycin also significantly reduced IL-1β in HPBCD-treated hRPE. These results provide evidence that the mTOR signaling pathway was also involved in SASP in cultured 7KC-treated or control hRPE.

### 2.4. Reduced Barrier Resistance in Senescent 7KC-Treated hRPE

To determine if 7KC-treated hRPE had reduced barrier resistance, the barrier integrity of hRPE was measured. Barrier structure was measured as the interactions between the adherence junction protein, cadherin, and β-catenin using co-immunoprecipitation (Co-IP) and by immunohistochemical labeling of junctional protein, ZO-1, in cell monolayers. Barrier function was measured using transepithelial resistance (TER) or cell impedance using electrical cell-substrate impedance sensing (ECIS). Compared to HPBCD, Co-IP of cadherin and β-catenin was significantly decreased in 7KC-treated hRPE without affecting the total cadherin and β-catenin concentrations (Figure 4A,B). The junctional integrity of the monolayer of the hRPE, assessed by immunolabeling for ZO-1, demonstrated intact cell-cell junctions in HPBCD-treated hRPE but disrupted junctions in 7KC-treated hRPE (Figure 4C), further supporting the idea that 7KC reduces RPE barrier integrity. To determine the effect of 7KC on barrier function, hRPE barrier resistance was measured by TER after 7KC and found to be significantly reduced compared to either untreated- or HPBCD-treated cells (Figure 4D). These findings were corroborated by ECIS in which 7KC treatment significantly reduced cell impedance compared to HPBCD treatment and pretreatment with rapamycin improved cell impedance in 7KC-treated hRPE as measured by ECIS (Figure 4E,F).

### 2.5. Serine Phosphorylation of IQGAP Regulates 7KC-Mediated RPE Senescence and SASP and Fibrosis of CNV Lesion

We previously found that serine phosphorylation of the multidomain protein, IQGAP1, was involved in TNFα-mediated RPE barrier compromise [25]. Therefore, we were interested in investigating whether IQGAP1 phosphorylation was involved in 7KC-mediated RPE changes, including 7KC-induced senescence. Since IQGAP1 phosphorylation is regulated by protein kinase C (PKC) [26], a series of experiments were performed in the hRPE pretreated with a pan PKC inhibitor, Go6983, prior to incubation with 7KC or HPBCD. Compared to HPBCD, 7KC treatment induced phosphorylation of PKC and IQGAP1 serine phosphorylation, measured by immunoprecipitation (Figure 5A). Pretreatment with Go6983 partially inhibited 7KC-induced p-PKC (Figure 5A) but greatly reduced IQGAP1 serine phosphorylation, supporting the hypothesis that PKC activation was upstream of IQGAP1 phosphorylation. In a parallel experiment, p21, VEGF, IL-1β, p-mTOR, and p-S6K1 were measured in Go6983 or control pretreated hRPE. 7KC-mediated p-mTOR, p-S6k1, p21, VEGF, and IL-1β were reduced in Go6983-pretreated cells (Figure 5B,C). Taken together, the results in Figure 5 show that 7KC-mediated hRPE senescence involved PKC and serine phosphorylation of the multidomain protein, IQGAP1.

We previously reported that intravitreal injections of 7KC promoted fibrotic changes of laser-induced CNV [15]. Since 7KC induces serine phosphorylation of IQGAP1, we then tested the effect of IQGAP1 serine phosphorylation on 7KC-mediated fibrosis in laser-induced lesions in mice. The serine residues 1441 and 1443 have been identified as important phosphorylation sites on human IQGAP1 [26]. Therefore, the serine residue 1441 (murine analog) on IQGAP1 was targeted as an area of interest important in AMD and fibrosis. We generated mice that had a serine 1441 mutation using a CRISPR-gene editing approach (CRISPR/IQ mice). Two weeks after laser-induced injury and intravitreal 7KC injections, we measured lectin- and αSMA-stained lesions in CRISPR/IQ and littermate wildtype mice. As shown in Figure 6, volumes of lectin-stained lesions were not significantly different between 7KC-injected eyes of littermate wildtype and CRISPR/IQ mice. However, CRISPR/IQ mutant mice had reduced αSMA-stained lesions after 7KC injection compared to littermate control mice (Figure 6). Taken together, the results support the postulate that IQGAP1 phosphorylation was involved in pathophysiologic steps related to 7KC-mediated fibrosis.

## 3. Discussion

AMD is a complex disease involving genetic and external stresses [27] and is believed to involve dysregulated signaling from oxidative stress and pro-inflammatory and angiogenic factors [28,29]. AMD also involves the effects of multiple cell types in the retina, including Műller cells, microglia, pericytes, RPE, and CECs, as examples. Although anti-VEGF agents have tremendously improved outcomes in neovascular AMD, fibrosis remains a leading cause of the failure of anti-VEGF agents in neovascular AMD [30]. 

Despite the involvement of many different cell types in AMD, much interest has focused on the dysfunction or damage to the RPE that precedes photoreceptor degeneration and vision loss [31]. RPE dysfunction occurs with various cellular events, including apoptosis, necrosis, and impaired autophagy [32]. In this study, we found that 7KC, an oxidized cholesterol that accumulates in drusen in AMD [4] causes RPE to become senescent and not undergo EMT or death. In contrast, 7KC caused mesenchymal transition in pericytes, Műller cells, and CECs [15] after exposure to the same culture conditions. 

Senescent cells accumulate with increased age and in age-related diseases [33]. However, premature senescence develops in diseases in response to stresses, including increased oxidative compounds and inflammation, which are implicated in AMD pathogenesis [29,34,35,36,37]. A study of cultured human RPE from donor eyes reported that the oxidative stressor, hydrogen peroxide (H_2_O_2_), and bone morphogenetic protein-4 interacted to promote the senescence of RPE [38]. In general, senescent cells adopt a complex phenotype characterized by permanent cell cycle arrest, increased β-galactosidase activity, resistance to apoptosis, and the adoption of SASP [20]. Increased p16 and p21, and decreased Lamin B1 are characteristic markers of senescent cells [39]. We found that 7KC-treated hRPE expressed p16 and p21 and had increased β-galactosidase activity and reduced LaminB1. In addition, 7KC-treated hRPE developed SASP with increased p-NF-κB, VEGF, and pro-inflammatory factors, IL-1β and IL-6, as well as activated mTOR, which has been shown to be important in RPE senescence [23]. Inhibition of mTOR signaling by rapamycin at a low dose inhibited 7KC-induced p-Histone H3, p16, p21, p-NF-κB, IL-1β, IL-6, and VEGF, providing evidence that 7KC induces RPE senescence and SASP through mTOR signaling. Our findings align with other studies in which inhibition of mTOR by rapamycin prevented SASP [40]. We also found that 7KC reduced RPE barrier function measured by TER and ECIS and that inhibition of mTOR signaling with rapamycin restored impedance. Targeting mTOR activation was tested in the past as a treatment for AMD [23,41] but did not reduce advanced forms of AMD in clinical studies [42]. Therefore, we sought other targets for potential treatment strategies.

We previously found that the multidomain protein, IQGAP1, when phosphorylated at a serine site, interfered with RPE integrity following exposure to TNFα [25]. In the current study, we assessed the role of IQGAP1 in 7KC-induced senescence in RPE related to the pathophysiology of AMD. PKC is an upstream regulator of IQGAP1 [26]. We found that a pan-PKC inhibitor reduced serine phosphorylation of IQGAP1 and p-mTOR, p21, VEGF, and cytokines that had been increased by SASP in 7KC-treated cultured hRPE. Serines 1441 and 1443 in IQGAP1 are important in cytoskeletal regulation [26]. Therefore, we developed mice with a point mutation in the murine analog (1441) of the human serine site of IQGAP1 using CRISPR-guided gene editing. We previously found that 7KC injection led to increased lectin-stained lesions and fibrosis determined by αSMA stains two weeks after laser injury in wild-type mice [15]. Therefore, we measured lectin- or αSMA-stained lesions two weeks after laser-induced injury and 7KC injection. We found no significant difference in lectin-labeled lesions between littermate wildtype and CRISPR/IQ mutant mice. This was not unexpected as neovascularization occurs earlier and regresses when fibrosis ensues in the laser-induced model [43]. There was a significant reduction in volumes of αSMA-stained lesions in CRISPR/IQ mutant mice compared to littermate controls after 7KC injection and laser injury. Taken together, these data support the hypothesis that IQGAP1 serine phosphorylation is involved in the formation of fibrotic lesions in the outer retina in neovascular AMD and that 7KC plays a role in RPE by inducing SASP with cytokine and VEGF release that activate and attract CECs to migrate toward the RPE. Once CECs make contact with 7KC in drusen [4], they are activated to transition to become invasive mesenchymal cells and migrate across the 7KC-induced compromised RPE barrier into the neural retina leading to vision loss. It remains unknown how 7KC enters cells to cause biological effects. 7KC may not directly trigger membrane receptor-mediated signaling but can increase membrane stiffness due to disoriented lipid packing [44,45]. This may be a mechanism whereby 7KC affects RPE barrier integrity and modulates signaling in the RPE.

Our findings enable us to revise our hypothesis of fibrosis in neovascular AMD. 7KC in drusen [4] reduces RPE barrier integrity and leads to SASP in RPE. Released cytokines and VEGF from the RPE activate CECs to migrate toward the 7KC in drusen associated with the basal RPE and undergo endothelial-mesenchymal transition, thereby becoming invasive cells with access to the neural retina through the compromised RPE barrier. Once fully transitioned, the mesenchymal-CECs lose their VEGFR2 markers and become resistant to anti-VEGF treatment. 

## 4. Materials and Methods

### 4.1. Animals and Procedures

Six-week-old C57BL/6J male and female mice were purchased from the Jackson Laboratory (Bar Harbor, ME, USA). All animal procedures were performed according to the Guide for the Care and Use of Laboratory Animals of the University of Utahand the Association for Research in Vision and Ophthalmology Statement for the Use of Animals in Ophthalmic and Vision Research and approved by IACUC and the Institutional Biosafety Committee of the University of Utah. Ketamine (100 mg/kg) and xylazine (20 mg/kg) were used for animal anesthesia, and cervical dislocation was performed for animal euthanasia under anesthesia. 

### 4.2. Cell Culture and Treatments

hRPE purchased from Lonza (Walkersville, MD, USA) were grown in retinal pigment epithelial cell growth media (RtEGM) and used from passages 3–5. hRPE purchased from ATCC (ARPE-19, Manassas, VA, USA) were grown in DMEM/F12 (Thermo Fisher Scientific, Waltham, MA, USA) media supplemented with 10% FBS and used up to 4 passages. Experiments performed with ECIS demonstrated that control-treated ARPE-19 or Lonza hRPE had similar resistances (2000–2200 ohms·cm^2^). hRPE was treated with 7-ketocholesterol (7KC, 10 µM, Avanti Polar Lipids, Birmingham, AL, USA) or volume-matched solvent control, 2-hydroxypropyl-β-cyclodextrin (HPBCD, MilliporeSigma, St. Louis, MO, USA) for 24 or 48 h. In some experiments, the hRPE was pretreated with rapamycin (10 nM, Selleckchem, Pittsburgh, PA, USA), Go6983 (10 μM, Tocris Bioscience, Minneapolis, MN, USA), or an equal volume of vehicle control DMSO for 30 min prior to incubation with 7KC or HPBCD.

### 4.3. Cell Senescence Measurement by β-Galactosidase Staining

A beta-galactosidase (β-gal) staining kit (Cell Signaling Technology, Danvers, MA, USA) was used to measure cell senescence. After treatment, hRPE was rinsed with PBS and then fixed with 1× Fixative Solution for 10 min at room temperature. After fixation and two washes with PBS, cells were incubated with Staining Solution at 37 °C for 2 h. Five–ten random fields in each well were imaged under a fluorescence microscope (Olympus, Tokyo, Japan). β-gal-stained positive cells were counted. 

### 4.4. RNA Extraction and Quantitative Real-Time PCR

Immediately following treatment, hRPE in 6-well plates were washed once with ice-cold 1× PBS and lysed in buffer RLT with β-mercaptoethanol (1:100, Sigma-Aldrich, St. Louis, MO, USA). Lysates were collected for RNA extraction using the RNeasy Kit (Qiagen, Valencia, CA, USA). 

Isolated RNA samples were reverse transcribed to cDNA using the High-Capacity cDNA Reverse Transcription Kit (Thermo Fisher Scientific). The cDNA of each sample was evaluated for the expression of specific genes using the TaqMan Gene Expression Master Mix and TaqMan probes against the genes of interest (Thermo Fisher Scientific). The ΔCT values for the specific genes were calculated using β-actin as the house-keeping gene control, and the 2^−ΔΔCT^ was calculated relative to the experimental control group.

### 4.5. Immunoprecipitation and Western Blots

After treatment, cells were lysed in a radioimmunoprecipitation assay buffer (RIPA) containing a protease inhibitor cocktail (Roche Diagnostics, Indianapolis, IN, USA) and a phosphatase inhibitor (ThermoFisher Scientific). Cell lysates were centrifuged at 1.6 × 10^4^ g for 5 min at 4 °C, and the protein concentration in the supernatant was quantified with BCA assay (Thermo Fisher Scientific). The cell lysate supernatant containing 300 mg of total protein was incubated with antibody to pan-cadherin (1:100, Cell Signaling Technology, Danvers, MA, USA) or IQGAP1 (1:100, BD Transduction Laboratories, Franklin Lakes, NJ, USA) and 10 µL Dynabeads protein G (Thermo Fisher Scientific) by gently rotating at 4 °C overnight. The antibody/protein/Dynabeads protein G complex was washed three times in RIPA buffer, resuspended in 2× sample buffer (Thermo Fisher Scientific), and denatured at 95 °C for 5 min to generate samples that were analyzed by western blot.

To measure protein expression in hRPE, 8–10 µg of protein from the cell lysate was loaded into NuPAGE 4–12% Bis-Tris gels (Thermo Fisher Scientific), transferred to a PVDF membrane, and incubated with antibodies against FAP (cat# 66562), alpha-SMA (cat#19245), BCL2 (cat# 2827), p16 (cat # 80772), p21( cat# 2947), LaminB1(cat# 17416), phospho-NF-κB (Ser536) (cat# 3039), NF-κB (cat# 4882), phospho-mTOR (Ser2448) (cat# 5536), mTOR (cat# 2972), phospho-p70 S6 kinase 1 (Thr389) (cat# 97596), p70 S6 kinase 1 (cat# 9202), pan-cadherin (cat# 4068), β-catenin (cat# 2677), IL-6 (cat# 12153), caspase 1 (cat #, 3866), phosphor-PKC (pan) (cat# 9371), PKCα (cat# 2056) ((1:1000, Cell Signaling Technology), VEGF (cat# sc-7269) (1:500, Santa Cruz Biotechnology, Dallas, TX, USA), BEST1 (cat# ab 2182), RPE65 (cat# ab175936), CRALBP (cat# ab182573), PDGFRβ (cat# ab69506), integrin β4 (cat# ab133682), IL-1β (cat# ab216995), or to phospho-Histone H3 (Ser10 cat# ab5176) (1:1000, Abcam, Cambridge, MA, USA), IQGAP1 (cat# 610612), and phospho-serine (cat # 612549, BD Biosciences, San Jose, CA, USA). All membranes were re-probed with HRP-conjugated β-actin (1:3000, cat# sc-47778, Santa Cruz Biotechnology, Dallas, TX, USA) to ensure equal protein loading. Densitometry analysis was performed with the use of the software UN-SCAN-IT version 7.1 (Silk Scientific, Orem, UT, USA). 

### 4.6. RPE Cell Death Measured by TUNEL Staining

Cell death was detected by an In Situ Cell Death Detection Kit, TMR Red (Sigma Aldrich) in hRPE treated with HPBCD or 7KC. After 48 h of treatment, hRPE grown on cover glasses (ThermoFisher) were first fixed in 4% paraformaldehyde for 15 min at room temperature. After two washes with PBS, cells were then incubated in freshly prepared permeabilization solution (0.1% Triton X-100 in 0.1% sodium citrate) for 2 min on ice. Following two washes in PBS, cells were incubated with a TUNEL-reaction mixture in a humidified incubator for 60 min at 37 °C in the dark. Cover glasses were then mounted with DAPI-Fluoromount-G (SouthernBiotech, Birmingham, AL, USA) after three washes with PBS. TUNEL-positive cells were detected under a confocal microscope (FV1000, Olympus, Shinjuku City, Japan) at 580 nM, and 5 images/samples were taken at 10× magnification for analysis.

### 4.7. hRPE Viability Assay 

Cell viability was measured by the Vybrant MTT Cell Proliferation Assay Kit (Thermo Fisher Scientific). hRPE grown in 96-well plates at a density of 10,000 cells/well were treated with HPBCD or 7KC (10 µM). Forty-eight hours after treatment, the medium was replaced with 100 µL of fresh medium/well, and 10 µL of the 12 mM MTT stock solution was added into each well. The microplates were then incubated at 37 °C for 4 h. After labeling the cells with MTT, 100 µL of the SDS-HCL solution was added to each well. The microplates were then incubated at 37 °C for another 4 h in a humidified chamber, and absorbance at 570 nm was read in a microplate reader (BioTek, Winooski, VT, USA).

### 4.8. Immunostaining of Junctional Protein ZO-1 in hRPE 

hRPE was cultured on cover glasses until a monolayer was formed. Cells were treated with HPBCD or 7KC (10 µM) for 48 h. After treatment, cells were fixed in 4% paraformaldehyde for 15 min, blocked in 5% normal goat serum in PBS/0.3% Triton X-100 for 1 h at room temperature, incubated with rabbit anti-ZO-I (1:200) overnight at 4 °C and incubated for 1 h at room temperature with Cyanine Cy^TM^3-conjugated goat anti-rabbit IgG antibody (1:200, Jackson ImmunoResearch Lab, West Grove, PA, USA). The cover glasses were then washed in PBS three times and mounted in DAPI-Fluoromount-G (SouthernBiotech, Birmingham, AL, USA). Images were captured using a confocal microscope (FV1000, Olympus, Japan).

### 4.9. Transepithelial Resistance (TER) and Electrical Cell-Substrate Impedance Sensing (ECIS) of RPE Barrier Resistance

RPE barrier resistance was monitored by TER (World Precision Instruments, Sarasota, FL, USA) and ECIS (Applied Biophysics, Troy, NY, USA). For TER measurements, hRPE was plated on 12-well (0.4 µm pore size Polyester PET) Transwell inserts (Corning, Corning, NY, USA). TER was monitored daily using an Endohm-12 Transwell Chamber connected to an EVOM Volt/ohm meter according to the manufacturer’s instructions until the TER reading was stable in each well. Cells were then incubated with HPBCD or 7KC (10 µM). TER was measured at 48 h after treatment.

ECIS was used to measure the resistance and impedance of the cultured hRPE barrier as described previously [25,46]. Briefly, 8WE10+ ECIS culturewares were coated with attachment factor (cell systems) for 30 min followed by electrode stabilization of all wells. 1.0 × 10^5^ hRPE or ARPE-19 cells were then seeded in each well and capacitance was monitored at 64,000 Hz. Once confluent, indicated by a capacitance value of ~5 nF, hRPE was pretreated with the mTOR inhibitor, rapamycin (10 nM), or volume-matched vehicle control, DMSO, for one hour followed by incubation with 7KC (10 µM) or volume-matched solvent control, HPBCD. Transepithelial electrical resistance (TER) was measured at 4000 Hz for each well for at least 24 h. For each well, normalized resistance was calculated by dividing the resistance at each time point by the initial resistance prior to the addition of reagents. Relative resistance was calculated by dividing the mean normalized resistance of the experimental groups by the mean normalized resistance of the control group. 

### 4.10. Generation of the IQGAP1 Serine 1441A Mouse Line 

The serine site 1441 is a known phosphorylation site on human IQGAP1 [26]. The transgenic core at the University of Utah produced the IQGAP1 S1441A/1441A mutant (herein referred to as CRISPR/IQ, registered with MGI and available following publication). CRISPR Cas9 reagents were designed by the University of Utah Mutation Generation and Detection Core. The sgRNA N20 sequence used was 5′-AGGTTGTTATCCTCCTTCAT-3′ and the single-stranded oligo deoxyribonucleic acid (DNA) nucleotide (ssODN) HDR donor sequence was 5′-CTATCCGCGATGCCAAAACCCCTGACAAGATGAAAAAAgCAAAGCCCATGAAGGAaGAcAACAACCTCAGCCTCCAGGAGAAGAAAGAGAAGATCCAGAC-3′. Base pair changes are in lower case. The ssODN contained stabilizing 5′ and 3′ phosphorothioate modifications, a single base change to introduce the S114A mutation, and two silent changes to block CRISPR cutting after HDR and to introduce a unique restriction enzyme site for genotyping. The University of Utah Transgenic and Gene Targeting Core co-electroporated a ribonucleoprotein complex and the ssODN donor molecule into single-cell embryos harvested at day 0.5. Electroporated embryos were rinsed and surgically implanted into oviducts of 0.5-day pseudopregnant females. Founders were genotyped with simple PCR and restriction fragment length polymorphism (RFLP) analyses. Founders with insertion were bred and the resulting N1 mice were sequenced to confirm correctness. Male and female mice on a C57Bl/6J background were used and routinely tested for *Rd1*, *Rd8,* and *Gnat2* mutations. A schematic of PAM sites for CRISPR/Cas9 mice generation was provided in Appendix A. All protocols were in compliance with AALAC procedures and approved by the IACUC committee.

### 4.11. Intravitreal Injections of 7KC and Laser-Induced Injury

Six-week-old homozygous CRISPR/IQ and littermate control male and female mice received bilateral injections of 7KC (Avanti Polar Lipids Inc., Birmingham, AL, USA), as previously described [15]. One week after intravitreal injections, the laser-induced injury was performed using the Phoenix Image-Guided Laser System 94 (Phoenix Micron IV, Pleasanton, CA, USA). Each eye received four laser spots at approximately two-disc diameters from the optic nerve. After laser treatment, the second intravitreal injection of 7KC was administered at a separate site from the first injection. The animals were euthanized 2 weeks after laser injury and the eyes were removed for choroidal flat mount analysis. 

### 4.12. RPE/Choroidal Flat Mount Preparation, Staining, and Lesion Volume Quantification

Freshly enucleated eyes were first fixed in 4% paraformaldehyde (Electron Microscopy Sciences, Hatfield, PA, USA) for 1 h and followed by an additional 1 h after removing the cornea, lens, vitreous, and retina. The posterior eyecups were then blocked in phosphate-buffered saline (PBS) containing 1% bovine serum albumin (BSA) and 0.5% Triton X-100 at room temperature, and followed by staining with AlexaFluor 568-conjugated Isolectin B4 (1:500, Invitrogen, Carlsbad, CA, USA), and αSMA (1:200, Cell Signaling Technology) overnight at 4 °C. To label αSMA staining, the eyecups were stained with FITC-conjugated goat anti-rabbit secondary antibody (1:200, Invitrogen) for 1 h at room temperature. After three washes, the eyecups were flattened by cutting incisions radially and mounted onto a microscope slide with a VECTASHIELD mounting medium (Vector Laboratories, Burlingame, CA, USA). 

Confocal images of each lectin or αSMA-labeled lesion with the z-stack set at 1 nm intervals were captured at 488 and 568 nm using a 20× objective on a Confocal Laser Scanning Microscope (Olympus Corporation, Japan). Images were imported into IMARIS (Oxford Instruments, Switzerland), and lesion volumes were determined using the Surfaces Module (Version 9.1.2, Bitplane, Santa Barbara, CA, USA). Lesions with obvious hemorrhage or bridging CNV were excluded.

### 4.13. Statistics

Results are presented as mean ± standard error (SE). A *p*-value < 0.05 was considered statistically significant. For in vitro studies, each experimental group had an *n* = 6–9 from at least two experimental replicates. In an independent experiment, we used cells from the same donor eye and evenly distributed each experimental group across different cultureware. Therefore, the measured outcomes were analyzed with a mixed-effects linear regression model with only experimental replicates as a random effect using Stata-17 software (StataCorp LLC, College Station, TX, USA). For the in vivo laser-induced CNV study, we analyzed outcomes using a mixed effects linear regression model with lesions nested within the same eye and eyes nested within the same animal using CNV volume measurements as the continuous outcome variable.

## Figures and Tables

**Figure 1 ijms-24-10276-f001:**
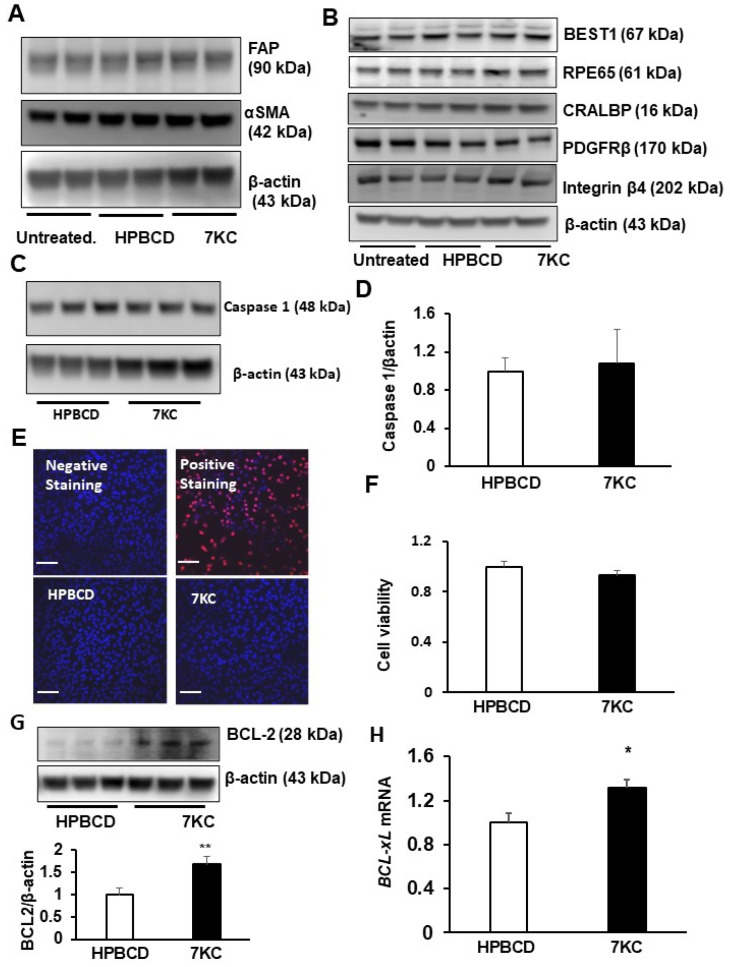
7-ketocholesterol (7KC) does not promote epithelial mesenchymal transition (EMT) or death of the retinal pigment epithelium (RPE). Western blots of (**A**) mesenchymal markers, fibroblast activation protein (FAP) and alpha smooth muscle actin (αSMA), (**B**) RPE specific markers, bestrophin 1 (BEST1), RPE65, CRALBP, PDGFRβ and integrin β4, (**C**,**D**) caspase 1 and densitometry; (**E**) TUNEL staining (the scale of the bars: 100 µm; red: tunnel positive cells), (**F**) hRPE viability measured by MTT assay, (**G**) western blot of BCL2, and (**H**) real time PCR of *BCL-xL* mRNA in HRPE treated with HPBCD or 7KC for 48 h or in untreated HRPE (* *p* < 0.05, ** *p* < 0.01 vs. HPBCD; *n* = 4–6).

**Figure 2 ijms-24-10276-f002:**
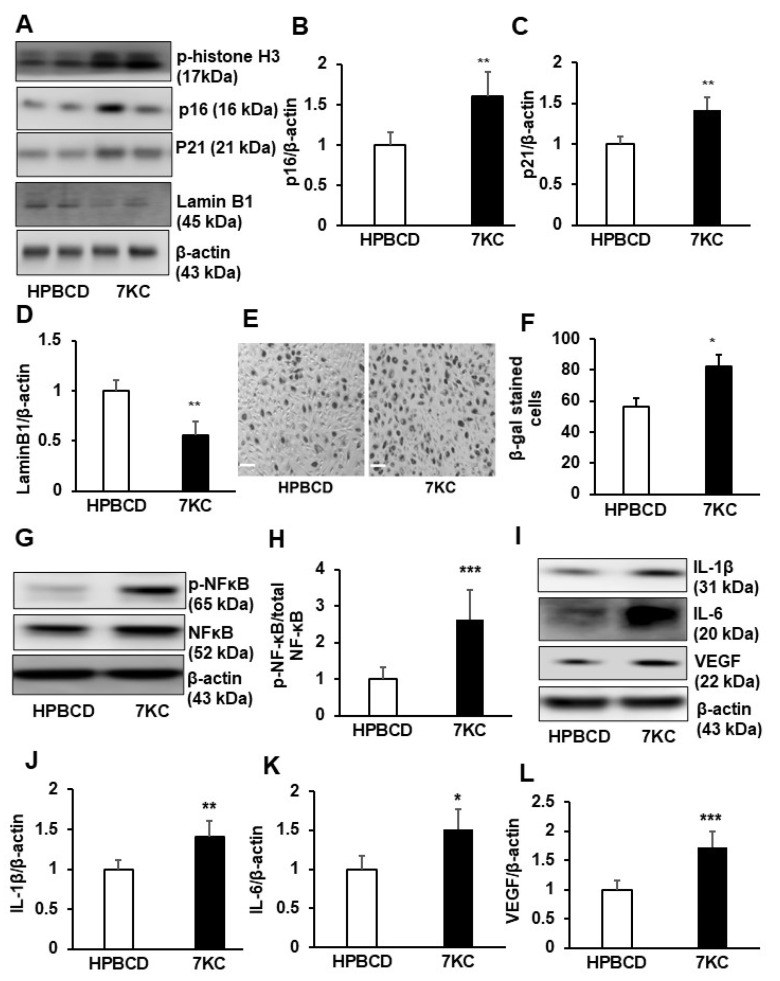
7-ketocholesterol (7KC) induces RPE cell cycle arrest at mitosis, senescence and senescence associated secretory phenotype (SASP). Western blots of (**A**) of phospho-Histone H3 (Ser10), (**A**–**D**) p16, p21 and LaminB1; (**E**,**F**) β-galactosidase (β-gal) (the scale of bars: 100 µm); western blots of (**G**,**H**) phosphorylation of NF-κB (p-NF-κB), total NF-κB, (**I**,**J**) IL-1β, (**I**,**K**) IL-6 and (**I**,**L**) VEGF in HRPE cells treated with HPBCD or 7KC for 48 h (* *p* < 0.05, ** *p* < 0.01, *** *p* < 0.001 vs. HPBCD; (**A**,**E**,**G**,**I**): representative images; (**B**–**D**,**F**,**H**,**J**–**L**): quantification results; *n* = 4–6).

**Figure 3 ijms-24-10276-f003:**
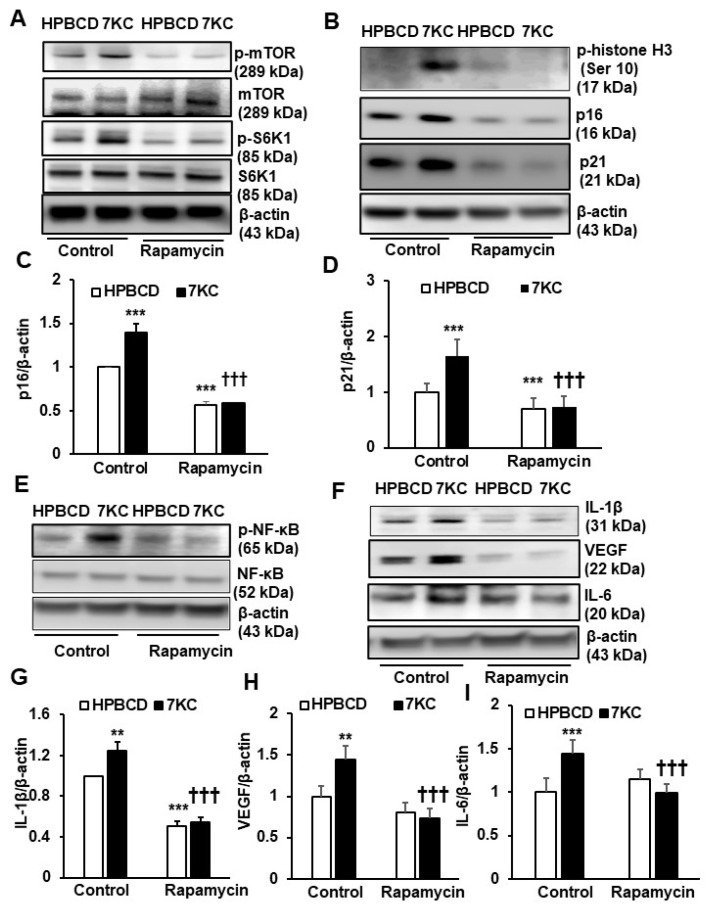
Inhibition of mTOR signaling inhibits 7-ketocholesterol mediated RPE senescence and SASP. Western blots of (**A**) phosphorylation of mTOR (p-mTOR), total mTOR, phosphorylation of S6K1 (p-S6K1), total S6K1, (**B**) phospho-Histone H3 (Ser10), (**B**,**C**) p16, (**B**,**D**) p21, (**E**) phospho-NF-κB (p-NF-κB) and total NF-κB, (**F**,**G**) IL-1β, (**F**,**H**) VEGF and (**F**,**I**) IL-6 in hRPE pretreated with Control or Rapamycin (10 nM) for 30 min prior to incubation with HPBCD or 7KC for 48 h (** *p* < 0.01 and *** *p* < 0.001 vs. HPBCD of Control; ^†††^
*p* < 0.001 vs. 7KC of Control; *n* = 4–6).

**Figure 4 ijms-24-10276-f004:**
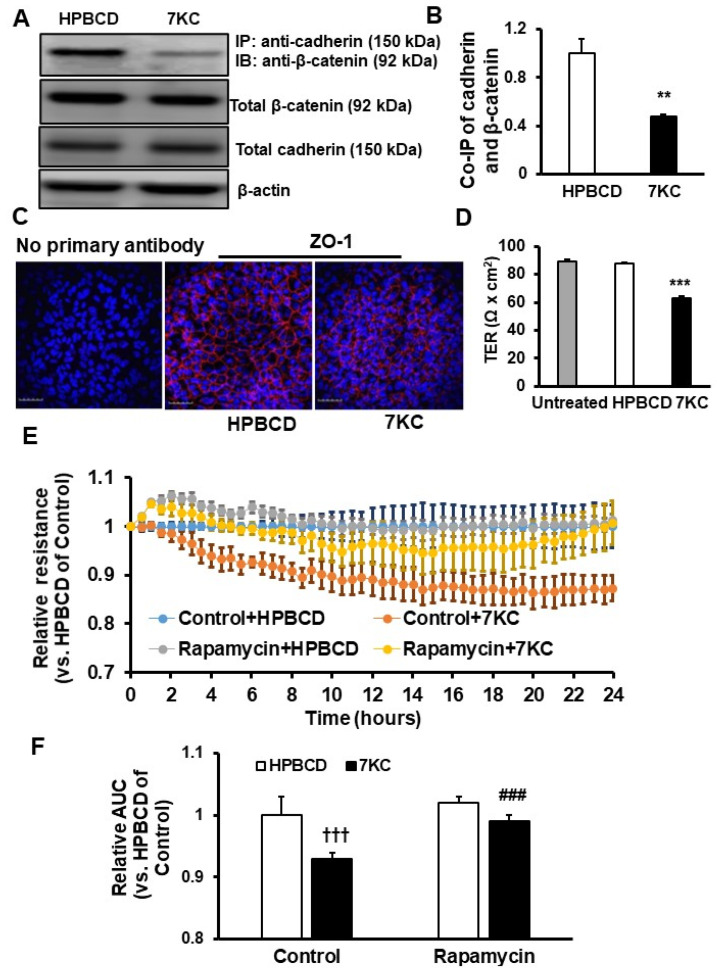
Inhibition of mTOR by rapamycin prevents 7-ketocholesterol (7KC)-mediated barrier compromise of RPE. (**A**,**B**): Co-immunoprecipitation of cadherin and β-catenin, (**C**) immunostaining of junctional protein ZO-1 (the scale of bars: 100 µm; red: ZO-1 stain), and (**D**) transepithelial resistance (TER) of hRPE treated with HPBCD or 7KC or without control or experimental treatment (untreated) for 48 h before being processed for different assays ((**A**), representative gel images, (**B**), quantification results; ** *p* < 0.01, *** *p* < 0.001 vs. HPBCD).; (**E**,**F**) Electric Cell-Substrate Impedance Sensing (ECIS) of hRPE treated with Control+HPBCD, Control+7KC, Rapamycin+HPBCD, Rapamycin+7KC ((**E**), relative resistance measurements and (**F**), quantification of area under the curve (AUC) of 24 h resistance measurements; ^†††^
*p* < 0.001 vs. Control+HPBCD, ^###^
*p* < 0.001 vs. Control+7KC; *n* = 4–6).

**Figure 5 ijms-24-10276-f005:**
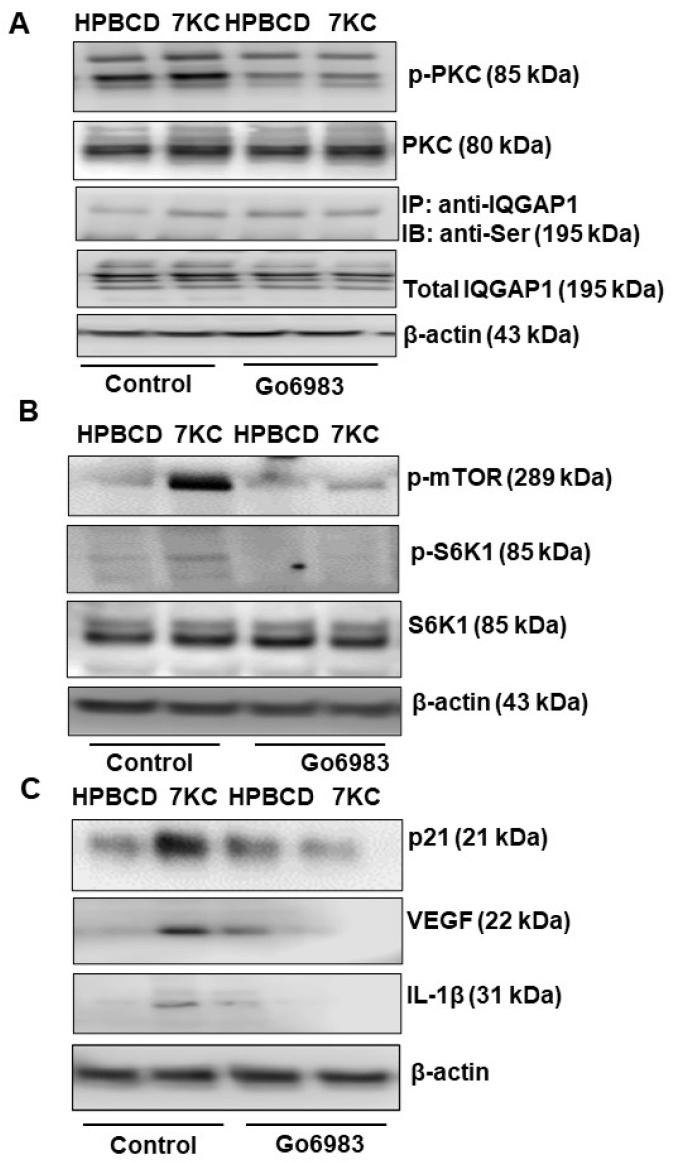
PKC-dependent serine phosphorylation is involved in 7KC-mediated RPE senescence and SASP. Western blots of ARPE-19 pretreated with control or Go6983 prior to incubation with HPBCD or 7KC and measured for (**A**) p-PKC, serine phosphorylation of IQGAP1 and total IQGAP1, (**B**) p-mTOR, total mTOR, p-S6K1 and total S6K1, and (**C**) p21, VEGF, and IL-1b (*n* = 4–6).

**Figure 6 ijms-24-10276-f006:**
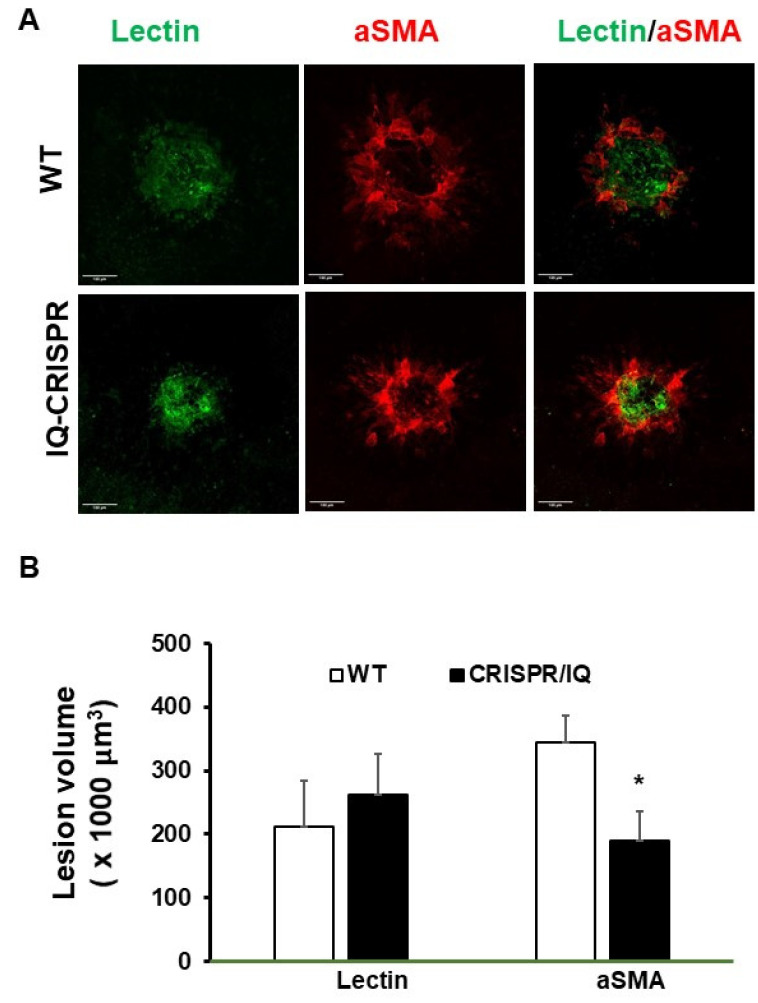
7KC promotes CNV fibrosis through IQGAP1 serine phosphorylation. (**A**) Representative choroidal flat mounts showing laser induced lesions labeled with lectin (Green) and αSMA (red) (the scale of bars: 100 µm) and (**B**) Lesion volumes measured by lectin or αSMA staining 2 weeks after laser and intravitreal 7KC injections in CRISPR/IQ vs. wildtype mice (* *p* < 0.05).

## Data Availability

The datasets generated and/or analyzed during the current study are available from the corresponding author upon reasonable request.

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
