# Peer review of "7-Ketocholesterol Promotes Retinal Pigment Epithelium Senescence and Fibrosis of Choroidal Neovascularization via IQGAP1 Phosphorylation-Dependent Signaling"

_ijms, 2023, doi:10.3390/ijms241210276_

Round 1

Reviewer 1 Report

Well written manuscript with conclusions supported by results. This reviewer also appreciates that un-cropped Western blots were provided. I have a few comments/suggestions

1] Fig 1: Missing information of each protein Molecular weight. Please add KDa information for each protein accordingly. Since no cleaved Caspase 3 was observed this can be removed from the figure.

2] Figs3,4,5: Missing information of each protein MW. Please add accordingly.

Figs6a and 4a: Scale bars are not clearly readable, please add in legend or increase front size of scale bars.

Materials and Methods: we the animals genotyped for rd1 and rd8 mutations? please mention.

WB information: Authors should add the specific catalogue/product number information for each antibody used.

FIg 5; WB images are out of focus. Re-acquire images/reproduce with higher quality.

CRISPR/Cas9 mice generation (schematic of PAM sites, gRNA) and N1 mice genotyping details should be provided in supplementary information.

Author Response

The Reviewer 1

General comment: “Well written manuscript with conclusions supported by results. This reviewer also appreciates that un-cropped Western blots were provided.”

Thank you!

Comment 1 “1] Fig 1: Missing information of each protein Molecular weight. Please add KDa information for each protein accordingly. Since no cleaved Caspase 3 was observed this can be removed from the figure.”

Response: We added the molecular weight of each protein in Fig.1 and removed cleaved caspase 3 (original Fig. 1C).

Comment 2: “2] Figs3,4,5: Missing information of each protein MW. Please add accordingly.”

Response:  We added the molecular weight of each protein accordingly for Figs3, 4, 5.

Comment 3: “Figs6a and 4a: Scale bars are not clearly readable, please add in legend or increase front size of scale bars.”

Response: The scale of the bars was added into the legends of Figs 6a and 4a, which is 100 µm.

Comment 4: “Materials and Methods: we the animals genotyped for rd1 and rd8 mutations? please mention.”

Response: Thank you. The genotyping information for rd1 and rd8 mutations was added into the Materials and Methods (Page 11): "Male and female mice on a C57Bl/6J background were used and routinely tested for Rd1, Rd8 and Gnat2 mutations."

Comment 5: “WB information: Authors should add the specific catalogue/product number information for each antibody used.”

Response: The catalogue for each antibody was included in the WB information.

Comment 6: “FIg 5; WB images are out of focus. Re-acquire images/reproduce with higher quality.”

Response: The WB images in Fig. 5 were replaced by ones with better resolution.

Comment 7: “CRISPR/Cas9 mice generation (schematic of PAM sites, gRNA) and N1 mice genotyping details should be provided in supplementary information.”

Response: A schematic of PAM sites for CRISPR/Cas9 mice generation was provided in Supplementary Figures.

Reviewer 2 Report

In this manuscript, the authors investigated the effects of 7KC on RPE and showed that 7KC increases senescence markers and proinflammatory factors in human primary RPE, and compromises RPE barrier function through mTOR signaling.  They also demonstrated that serine phosphorylation of IQGAP is involved in 7KC-induced RPE changes and 7KC-mediated fibrosis in a mouse model of laser-induced choroidal neovascularization. Together with their previously published data on the effects of 7KC on choroidal endothelial cells (CECs), they identified mechanisms by which 7KC, abundant in drusen, can affect both RPE and CECs and promote choroidal neovascularization and fibrosis. This manuscript is professionally written, the methodology is sound, and the results and discussion are relevant. I recommend its publication.

There are only a few minor issues:

1. Please indicate the scale of the bars in Fig1e, Fig2e, Fig4c, and Fig5a in the legend.

2. Please also indicate in the legend the “n” for each group in the column charts, and that “data are presented as Mean±SE”.

3. The gene name of BCL-xL in Fig1h should be italicized.

4. The original gel images especially for Fig2 are difficult to read, I can barely find the bands used in the manuscript. For example, in the gel images for p-Histone H3, Lamin-B1, p-NFkB, VEGF, which band/subunit was used for quantification and present in the figure?

Author Response

Reviewer 2

General comment: “In this manuscript, the authors investigated the effects of 7KC on RPE and showed that 7KC increases senescence markers and proinflammatory factors in human primary RPE, and compromises RPE barrier function through mTOR signaling.  They also demonstrated that serine phosphorylation of IQGAP is involved in 7KC-induced RPE changes and 7KC-mediated fibrosis in a mouse model of laser-induced choroidal neovascularization. Together with their previously published data on the effects of 7KC on choroidal endothelial cells (CECs), they identified mechanisms by which 7KC, abundant in drusen, can affect both RPE and CECs and promote choroidal neovascularization and fibrosis. This manuscript is professionally written, the methodology is sound, and the results and discussion are relevant. I recommend its publication.”

Thank you!

There are only a few minor issues:

Comment 1. “Please indicate the scale of the bars in Fig1e, Fig2e, Fig4c, and Fig5a in the legend.”

Response: The scale of bars in Figs 1e, 2e, 4c and 6a were added in the figure legends, which is 100 µm.

Comment 2. “Please also indicate in the legend the “n” for each group in the column charts, and that “data are presented as Mean±SE”.”

Response: “n” for each group was indicated in the legends

Comment 3. “The gene name of BCL-xL in Fig1h should be italicized.”

Response: BCL-xL in Fig 1h is now in italics.

Comment 4. “The original gel images especially for Fig2 are difficult to read, I can barely find the bands used in the manuscript. For example, in the gel images for p-Histone H3, Lamin-B1, p-NFkB, VEGF, which band/subunit was used for quantification and present in the figure?”

Response: The bands used for quantification and present in the figures were labeled in revised original gel images.

Round 2

Reviewer 1 Report

Authors have addressed all of my comments in their revised manuscript.

One minor comment, Fig. 5A (b-actin), please crop this image accordingly, an additional protein band is visible in this image.